# Quantum spin models for numerosity perception

**Jorge Yago Malo**[ID][1‡]*, **Guido Marco Cicchini**[2‡], **Maria Concetta Morrone**[3], **Maria Luisa Chiofalo**[ID][1]

**1** Department of Physics "Enrico Fermi" and INFN, University of Pisa, Pisa, Italy, **2** Institute of Neuroscience, CNR-Pisa and PisaVisionLab, Pisa, Italy, **3** Department of Translational Research and of New Surgical and Medical Technologies, University of Pisa and PisaVisionLab, Pisa, Italy

‡ JYM and GMC are co-first authors on this work.
* jorge.yago@unipi.it

**Data Availability Statement:** The data underlying the results presented in the study are available from: DOI 10.5281/zenodo.7692492.

**Funding:** JYM was supported by the European Social Fund REACT EU through the Italian national

## Abstract

Humans share with animals, both vertebrates and invertebrates, the capacity to sense the number of items in their environment already at birth. The pervasiveness of this skill across the animal kingdom suggests that it should emerge in very simple populations of neurons. Current modelling literature, however, has struggled to provide a simple architecture carrying out this task, with most proposals suggesting the emergence of number sense in multi-layered complex neural networks, and typically requiring supervised learning; while simple accumulator models fail to predict Weber's Law, a common trait of human and animal numerosity processing. We present a simple quantum spin model with all-to-all connectivity, where numerosity is encoded in the spectrum after stimulation with a number of transient signals occurring in a random or orderly temporal sequence. We use a paradigmatic simulational approach borrowed from the theory and methods of open quantum systems out of equilibrium, as a possible way to describe information processing in neural systems. Our method is able to capture many of the perceptual characteristics of numerosity in such systems. The frequency components of the magnetization spectra at harmonics of the system's tunneling frequency increase with the number of stimuli presented. The amplitude decoding of each spectrum, performed with an ideal-observer model, reveals that the system follows Weber's law. This contrasts with the well-known failure to reproduce Weber's law with linear system or accumulators models.

## Introduction

Humans and animals possess the ability to estimate how many objects are present in a given space or in a time interval, that is their numerosity. Humans perform this task remarkably well, with an error rate of about 20% over a very large range of items, up to 200 [1–4].

Determining the number of elements in a set is traditionally associated with counting. However, it emerges as a perceptual function taking place even without attention and with higher precision than related perceptual functions (such as area and density estimation) [5, 6].

program PON 2014-2020, DM MUR 1062/2021.
GMC was supported by Horizon 2020 European
Research Council Advanced Grant GenPercept No.
832813 (to D.C.B.), Italian Ministry of Education
PRIN2017 Grants 2017SBCPZY, and FLAG-ERA
Joint Transnational Call 2019 Grant DOMINO.
MCM was supported by GenPercept ERC-adv no.
832813 and PRIN 2017.

**Competing interests:** The authors have declared
that no competing interests exist.

Several neurophysiological studies have demonstrated that, in the primate, brain neurons selective to numerosity do exist and their tuning is proportional to the preferred numerosity [7]. Remarkably, it is known that the task-associated uncertainty increases linearly with the numerosity itself, a law named after Weber [1]. Weber's law occurs regardless of the modality of presentation, i.e. whether numbers are delivered as distinct items in a single display, or as a sequence of items in the visual, auditory or tactile modality, as well as in the same position or in different ones. All these observations point to Weber's law as a hallmark of the central mechanism which estimates numerosity from different sensory formats. Indeed, Weber's law fails only under particular circumstances [3, 8], such as when items are too close together to be segregated, and mechanisms for estimation are based on the visual-texture grain and follow a square-root law. Interestingly, individual performance with sparse displays is predictive of mathematical literacy of the individual, whereas performance with high clutter display does not [9]. This suggests that the perceptual mechanisms for rapid estimation of sparse displays are core to mathematical knowledge, whereas those that enable perception in high clutter displays are not.

It is as well interesting that Weber's law cannot be attained by the simplest model of numerosity perception, which is one that counts the number of items in a region of space (or events in a period of time). In fact, the statistics of the latter follow a Poissonian distribution, implying that the variance scales with numerosity and hence that the standard deviation scales with its square root. This is a well-known problem in time perception where a central assumption is Poisson pacemaker: here, Weber's law has been modelled only resorting to ad-hoc assumptions, otherwise it resembles the signature behaviour with cluttered presentations [10, 11]. Currently, the neural mechanisms allowing Weber's law over such a large range of numerosities and conditions are not yet known.

Artificial Neural Network have been demonstrated to be a valid approach to the simulation of many aspect of perception. Pioneering work by John Hopfield, employing statistical mechanics, demonstrated that it was possible to study the behaviour of unsupervised networks comprised of simple processing units, based on spin-glass models in physics [12]. Interestingly, Stoianov and Zorzi using a network of this kind, successfully modelled human processing of numerosity [13]. The network, comprising one input layer and 2 hidden layers and trained in the classification of 2 dimensional stimuli containing various number of objects, spontaneously built a representation of numerosity, which could then be decoded by a simple linear classifier. If, in this model, the representations read-out of the classifier were used to perform a numerosity comparison, the behaviour would comply with Weber's Law.

More recently, Nasr et al. [14] have found that some units of a deep convolutional neural network (CNN) designed and trained for objects recognition, could also perform number discrimination on static displays. Using 8 convolutional layers for local filtering and 5 pooling layers for aggregated response, this CNN exhibited numerosity recognition in around 10% of its units, with an approximate Weber's law behaviour. Similar conclusions have been drawn with untrained networks [15], once again involving a limited amount of units. While these results are encouraging, a number of questions arise. In particular, whether numerosity perception is a global property of such neuronal network, since it is only encoded in a small percentage of a highly complex architecture. Another important problem involves the generalization to dynamic stimuli that unfold over space and time.

On the other hand, the last decade has seen the emergence of quantum models [16] to tackle non-linear complex dynamics. This Quantum-like paradigm [17] is based on the idea that the mathematical toolbox of quantum mechanics can describe phenomena which do not need to be quantum. Here, we note that while the classical models introduced by Hopfield [12] and our quantum mechanical model use similar spin Hamiltonians, see e.g. [18], the approaches

are substantially different. The classical models focus on using adaptable statistical structures [19, 20] aiming at defining a flexible network connectivity suited for learning models [21] to encode the input information. In contrast, our quantum spin model highlight the fact that minimal symmetric small-sized networks, with pre-determined connectivity, are enough to recover input information.

Examples of the use of quantum models include the study of the generation of persistent quantum-chaotic patterns at a microscopic scale and the amplification of quantum effects to a macroscopic scale have been investigated [16, 22]. Quantum versions of the IIT theory for consciousness [23, 24] and information processing [25] have also been proposed. The general features have been explored by using dissipative quantum models of the brain, driven by entanglement, and the corresponding use of quantum field theory to model brain functional activity [26]. Mathematical methods of quantum theory, especially quantum measurement theory, have been used to describe information processes in biosystems, and then applied to model combinations of cognitive effects and gene regulation [27]. Quantum-inspired techniques in machine learning have also been introduced in psychology [28] leading to the field of the so-called quantum cognition [29], hinging instead on the mathematical structure of quantum probability. This approach has been used to address cognitive phenomena, such as reinforcement learning during human decision-making [28], known to be hard to be described by means of classical probability theory [30]. For example, when incompatible events are involved, incompatibility produces superposition states of uncertainty which eventually result in violations of classical/probability laws [31]. Along these lines, it has been highlighted how, to some extent, the brain could work as a not only classical but also quantum Bayesian-inference machine [32, 33]. Moreover, several proposals for the use of quantum spin models and neuronal activity has been discussed [12, 34].

Such approaches do not necessarily imply the emergence of microscopic quantum effects in the brain. This timely area of research *per se* [35–37], has been recently revisited based on increasing experimental efforts [38, 39] also in view of proposed theories [40]. This flourishing research field is developing in parallel to advancements in the field of quantum biology [41], following on tremendous progress in experimental methods to investigate energy-excitation transfers in light-harvesting complexes [42].

Here, we use the nature of quantum-mechanics as a formal toolbox to describe the highly dynamical complexity underlying the encoding of numerosity perception. We present an example of a coarse-grained quantum model as a mathematical toolbox to map information processing of a network of neurons. We find that under general enough conditions, a quantum network with a very simple architecture is capable of counting the number of events as a global property, even when these excitations are introduced randomly in space and/or in time, with no need of any training and standard agnostic decoding.

## Methods

Our main goal is to design a successful model for numerosity perception as an intrinsic and very general tool for dynamical stimuli.

When dealing with a network of neurons and its mapping into a quantum model, a rather natural idea is to associate to each neuron an on-off state of activity which, in the quantum domain, can be described in terms of a spin 1/2. Assuming this as a starting point, our modelling is developed based on a minimal set of essential traits and functions that the network should possess, while yet guided by the observation that the numerosity perception manifests as a global response and an inherent property of the network, robust with respect to space and time randomness and dissipation. In the first column of Table 1 we provide the list of the

**Table 1. Minimal elements for information processing and their quantum counterpart.**

| Neural systems | Quantum physics model |
| --- | --- |
| Neuron/unit: activity vs no activity | Spin: orientation (up/down) |
| Propagation of activity | Excitation transfer via tunneling/exchange |
| Built-in connectivity of the network with excitatory and inhibitory mechanisms (receptive field/inhibitory surrounds) | Interaction potential between nodes |
| Flexibility in modeling connectivity of neighbors for different functions | From nearest-neighbor to all-to-all coupling |
| System decays to a resting state | Dissipation mechanisms |

Mapping of the phenomelogy in neuronal systems and the corresponding element in the quantum modelization.

essential neuronal functions we consider relevant to address numerosity perception. The second column in Table 1 reports instead the mapping onto the corresponding objects or functions of the quantum model.

## Quantum spin model

Based on the mapping in Table 1, we consider a 1D spin-1/2 chain, see Fig 1, with variable coupling ranging from simply nearest-neighbours (n.n.) to all-to-all (a.a.) connectivity. Excitations in the system, that we will model as ↑-polarised spins, can be transferred to connected sites via the exchange coupling. Moreover, the system is subject to an energy offset whenever there are several excitation in a near vicinity. The system Hamiltonian is then given by:

$$\hat{H} = \sum_{i,j}^{M} \left( J_{ij} \left( \sigma_i^+ \sigma_j^- + \text{h.c.} \right) + \Delta_{ij} (\sigma_i^z \sigma_j^z) \right),  \tag{1}$$

where $\sigma_i^{(x,y,z)}$ correspond to the Pauli matrices in site $i \in (1, M)$, $\sigma_i^{(+,-)}$ are the corresponding raising and lowering operators, $J_{ij}$ are the local exchange amplitudes between connected spins. Here we consider $J_{ij} = J = 1$ whenever two sites are linked and zero otherwise. This means for nearest neighbours $J_{ij} = 1$ if $|i - j| = 1$ and $J_{ij} = 1 \, \forall i, j$ for the all-to-all coupling. $\Delta_{ij}$ represent the

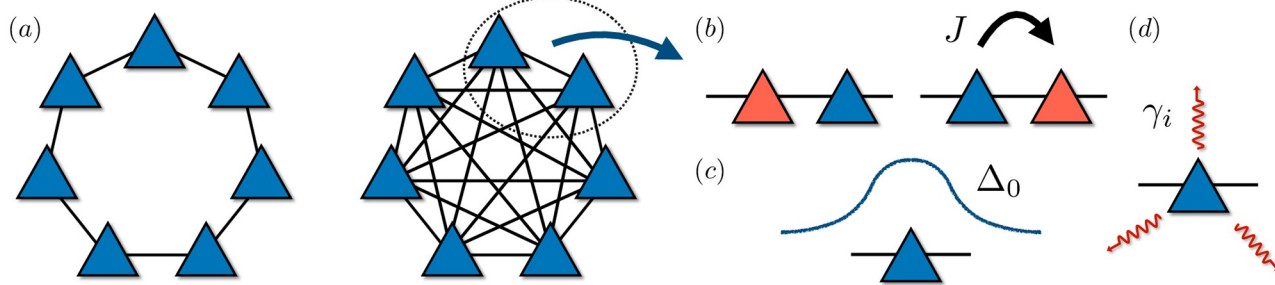

**Fig 1. Quantum spin model.** (a) Diagram of the spin system with variable connectivity, we present the case of nearest-neighbour and all-to-all coupling; (b) Diagramatic representation of the Hamiltonian and dissipative terms: each spin can propagate an excitation to its neighbours through the exchange term with amplitude *J*. Each spin experiences an energy offset given by the state of its neighbours. This energy shift is given by a gaussian profile centered in each spin with amplitude $\Delta_0$, (c). Finally, each spin can interact with different dissipative channels, (d) leading to excitation losses and dephasing with a rate $\gamma_i$.

offsite energy offsets defined as:

$$\Delta_{ij} = \Delta_0 \exp[-(i-j)^2/(2\sigma^2)] \,, \tag{2}$$

with $\sigma$ a constant that determines the width of the interaction potential between sites, and $\Delta_0$ the overall amplitude of such potential.

In the presence of coupling to an external environment, the dynamics of the system is described by its density operator $\rho(t)$, whose evolution is governed by the Gorini–Kossa-kowski–Sudarshan–Lindblad (GSKL) equation [43]:

$$\frac{d\rho}{dt} = -\frac{i}{\hbar}\left[\hat{H}, \rho\right] - \frac{1}{2\hbar}\sum_m \gamma_m (J_m^\dagger J_m \rho + \rho J_m^\dagger J_m - 2J_m \rho J_m^\dagger) \,, \tag{3}$$

where $J_m$ represents the $m$-th dissipation channel and $\gamma_m$ its corresponding dissipative rate. In our case, we consider two possible dissipative sources: excitation losses, here corresponding to spin flips and $J_m = \sigma_m^-$; and dephasing, due to collisional processes limiting the quantum coherence of the coarse-grained system model, here $J_m = \sigma_m^z$.

Note that the Hamiltonian we have introduced in Eq 1 is a paradigmatic model for quantum magnetism, introduced initially by Heisenberg [44] and first solved by Yang and Yang [45] and Baxter [46]. The model has a rich phase diagram exhibiting (anti)ferrogmatism both in the axial and planar axis. Continuous developments in both numerical and analytical tools have made the model relevant in both closed and dissipative scenarios up to these days, including for the simulation of systems unrelated to quantum magnetism as it is our case.

In order to produce the numerical simulations, we take advantage of stochastic unraveling methods, in particular quantum trajectories, allowing to reduce the numerical complexity of the computation at the cost of computing several trajectories. Details on the method can be found in [47].

We compute the time evolution of the system given by (3) for a series of relevant parameters, with the aim of building intuition on the expected dynamics in such a spin system.

In Fig 2, we present the evolution of the magnetization for an example system with $M = 7$ spins, where at the initial time $t = 0$ one of the spins is polarised in the $\uparrow$ direction while the rest remain in the resting state corresponding to $\downarrow$ orientation. As time progresses, the excitation can travel along the system due to the exchange coupling. In the nearest-neighbour (n.n.) case, shown in Fig 2a–2d, the excitation travels in a well-defined cone before covering the whole system and continues to travel back and forth as time evolves as we observe some revivals where the excitation returns to the original site after spreading across the whole system. The presence of the interaction $\Delta$, Fig 2b, modulates the profile of the evolution but does not change the overall behaviour. This is true for a wide range of interaction parameters $(\Delta_0, \sigma)$ as seen in Fig 2c where we present, as an example, a wider interaction profile. The addition of dissipation $\gamma_l$, Fig 2d, in the form of spontaneous spin decay uniformly reduces the signal in space and time, as predicted.

On the other hand, the all-to-all (a.a.) coupling scenario illustrated in Fig 2e–2h is remarkably different. We observe that the dynamics are much simpler. Here, the excitation is evenly spread into all the other sites—as they are all connected—to return back to the initial site at a rate $\propto J$, that is the tunneling time. There, we can interpret all the initially $\downarrow$ sites as a single "supersite" where the excitation is evenly transferred to each one of them, to subsequently return in this oscillatory manner. In the presence of interaction, Fig 2f and 2g, the rate of tunneling varies between the individual sites and, as a result, some non-uniform spreading is observed while time evolves. The role of dissipation is similar to the n.n. case, reducing the overall excitations in the system uniformly over time. Most of the highlighted features remain

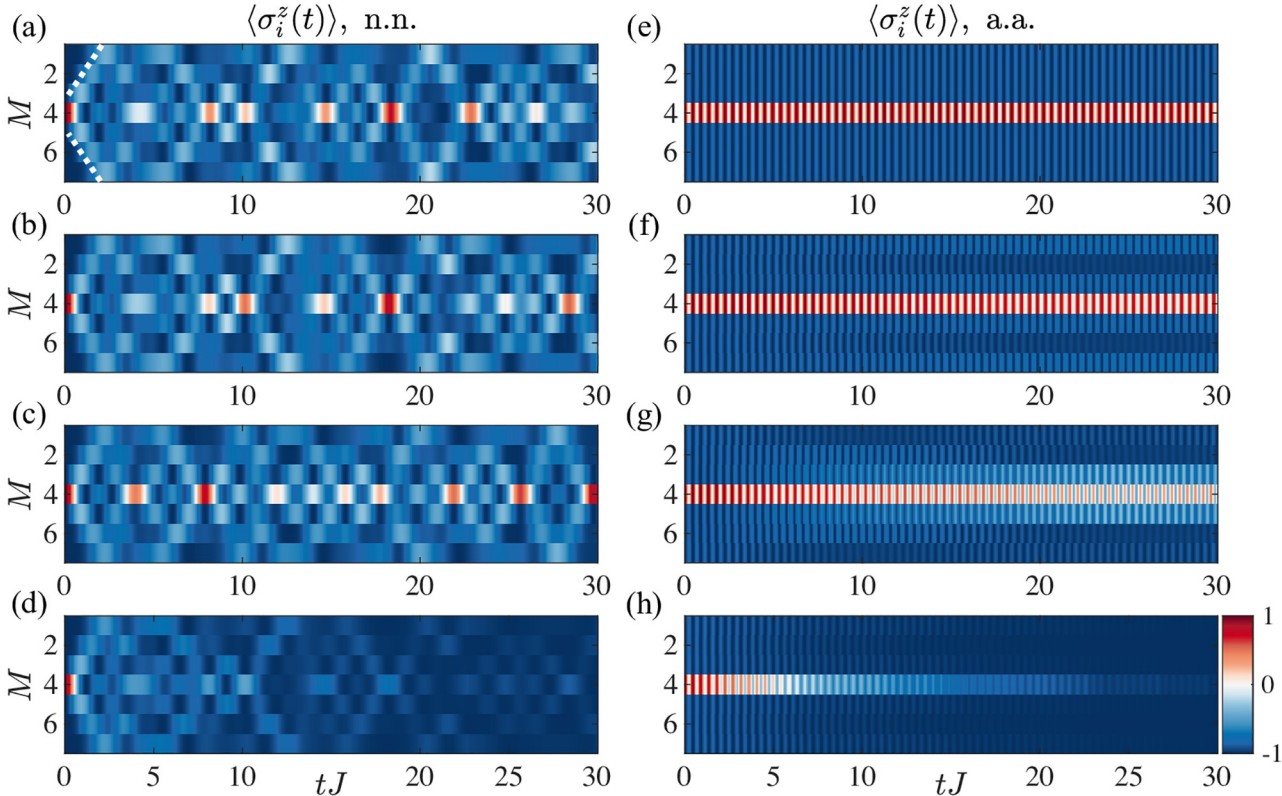

**Fig 2. Magnetisation profiles.** (a) Evolution of the local magnetisation $\sigma_i^z$ for a system with $M = 7$ sites, $J = 1$, $\Delta_0 = 0$, $\gamma_l = 0$ starting from a single excitation (spin up) on the middle site at $t = 0$ and nearest neighbor (n.n.) coupling. The dashed line highlights the magnetisation spreading in a light-cone manner; (b) same as (a) for $\Delta_0 = 0.1$, $\sigma = 1/\sqrt{2}$; (c) same as (a) for $\Delta_0 = 0.1$, $\sigma = 2/\sqrt{2}$; (d) same as (a) for $\gamma_l = 0.1$; (e)-(h) Same as (a)-(d) for the all-to-all (a.a.) case. In the n.n., we can observe that the excitation propagates in a light cone until reaching the boundaries, and then oscillates back and forth. The addition of interaction leads to space modulations that build over time changing quantitatively the magnetisation but without affecting the general behaviour. Inclusion of a spin decay rate $\gamma_l \neq 0$ leads to the loss of the excitation over time. In contrast, the a.a. does not display any excitation cone and the spin up evenly spreads to all the neighbours and bounces back and forth with a constant frequency, including interaction we observe again the appearance of space modulation in the propagation profile.

unchanged when we consider multiple excitations in the system. For the sake of completeness, we include several examples in the S1 Appendix, see Fig 7.

We have so far focused on the case of adding one single excitation to the system. However, the most interesting aspect of the model comes to play when we consider several spin flips along the time evolution. In particular, we observe that the power spectrum of the time signals contains information on the number of spin flips that took place during that evolution. In Fig 3, we consider the frequency spectrum of the magnetisation $\langle \sigma_i^z \rangle$ in a given site for a chosen time window of length $10 \leq tJ \leq 20$. In the initial time interval $t \leq 10J$ excitations in the form of spin flips in sites $i = 1, 3, 5$ respectively have been added to the system at fixed times. Here, we observe that in the n.n. case the spectrum has little dependence on the number $N$ of spin-↑ in the evolution; finding a set of narrow peaks at low frequencies that do slightly change in width and amplitude with no consistent dependence on $N$.

When the system is all-to-all coupled instead, we observe the appearance of peaks with equidistant frequencies for every additional spin flip. The temporal evolution of the magnetisation contains information on the numerosity for each site: a global increase of the number of excitations does increase the number of peaks with low-frequency harmonic components.

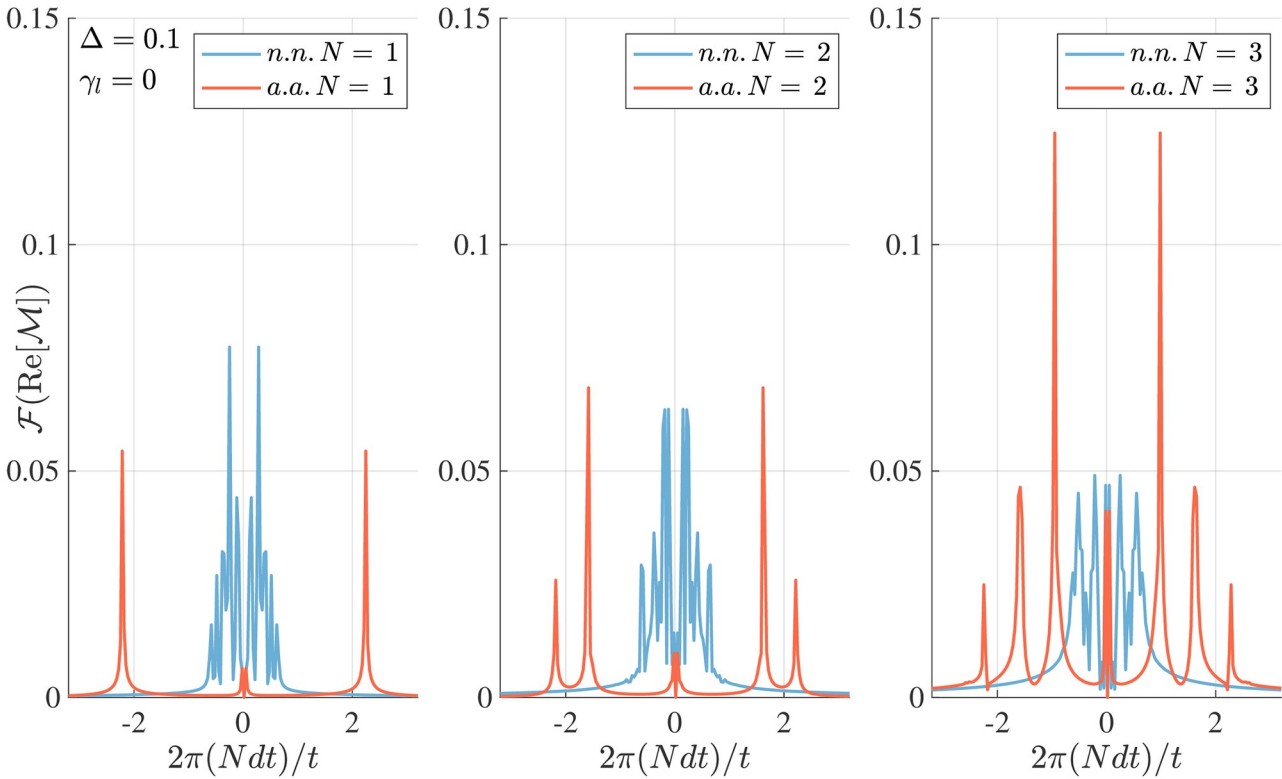

**Fig 3. Frequency spectrum of the magnetisation.** Amplitude spectrum of the magnetisation $\langle \sigma_i^z \rangle$ on site $i = 5$ as a function of time in a system with parameters $M = 7$ sites, $J = 1, \Delta_0 = 0.1, \sigma = 1/\sqrt{2}, \gamma_l = 0$ for varying number of spin flips during the evolution $N = 1, 2, 3$ displayed from left to right. In every panel we compare the results for the case of nearest-neighbours and all-to-all coupling, observing that the number of peaks and the overall behaviour of the spectrum shows small differences in the case of n.n., contrasting the a.a. case, where every time that a spin flip connects to a new magnetisation sector a new peak at a constant distance appears in the spectrum, showing the potential ability for the system to count. In both cases, the time domain used was $10 \leq tJ \leq 20$ after the events occurring in sites $i = 1, 3, 5$ respectively at fixed times in the initial window between $0 \leq tJ \leq 10$.

Note that due to the up-down symmetry of the model, we can only distinguish up to $N \leq M/2$ excitations, as an $N$ number of spin-↑ higher than $M/2$ can be regarded as simply $M - N$ spin-↓ particles.

We have presented here the case with no dissipation. However, the results are comparable if we consider any time window where the dissipation has not yet washed out all of the excitations from the system. We have also verified that the spectrum peaks do not depend on the specific time of, or interval between the measurements, nor on rotation angle, location of the spin flips, or the length of the time signal with the only variation being their relative amplitude, meaning that this is a robust feature. Details on these findings are in the Supporting information, see Figs 8 and 9 in S1 Appendix.

## Estimation protocol

As we discussed, Fig 3 shows that larger numerosity of inputs introduce lower frequency components in the temporal evolution of the magnetization. Thus, to validate that the information for all numerosity is adequately encoded in the magnetization profile, we applied an ideal observer model based on a supervised classifier. In Fig 4 we show an outline of our estimation protocol. In order to estimate numerosity in our system, we first record the local magnetisation, which is given by the expectation value of the observable $\langle \sigma_i^z \rangle$ on every site as a function

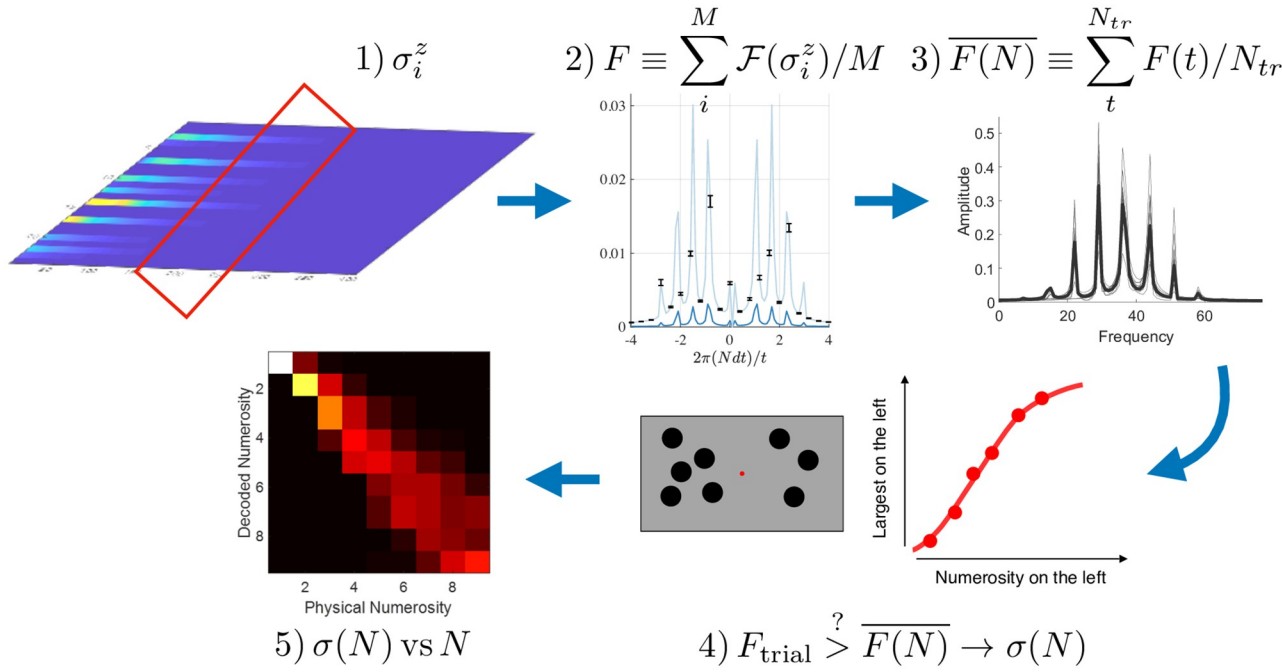

**Fig 4. Agnostic estimation protocol summary.** (1) The local time signal of the magnetisation is registered in a chosen time window, after the excitations have entered the system; (2) The spectrum of the signal is computed and then averaged over the number $M$ of sites in the network; (3) Averaging over realisations with excitations at random times, locations, and random angles produces a spectrum template for each numerosity; (4) Sample runs are then compared with the templates, using agnostic decoding; (5) The estimated errors are compared for different numerosities to assert whether the system follows the predicted Weber's law.

of time, where the profiles will resemble the patterns in Fig 2. We record the magnetisation in a fixed time window after all the stimuli have been added to the system, e.g. $10 \leq tJ \leq 20$ in Fig 3. This is displayed in Fig 4.1.

As we deliver stimuli randomly over space and time, we compute the spectrum from the local magnetisation at each site and then produce a space-average spectrum for every single trajectory.

In order to decode the information embedded in the power spectra, we analyze them through an ideal observer approach, that is agnostic to the signal having a quantum or classical origin. In this approach, the decoder assesses the similarity of an input spectrum to that of a library of spectra which have been previously learned, and informs the decoder on the average behaviour of the system in response to one of the possible inputs. This method requires no knowledge of the internal processes that give rise to the specific states from a given input. It simply classifies the current internal state, comparing it to a library of previously learned power spectra that are averaged over a fixed number of simulations of order $10^2$ (200 in the case of our results), each associated with a given numerosity, and eventually decodes it as the numerosity which on average produces the most similar internal state [48]. See Fig 4.3.

Once the system has learned the average behaviour for each numerosity, we consider an ideal decoder for the power spectrum of a new sample with a yet-to-be-decoded number of spin flips. The similarity is evaluated by correlating the spectrum of the new input signal with those present in the library representing the average spectrum to each numerosity. Then the template associated with the highest correlation is selected. All correlations are accepted and no threshold is used.

## Numerosity classification

In order to measure how the performance of the model varies as a function of the underlying numerosity, we run a virtual psychophysical experiment, as illustrated in Fig 4.4, in which two numerosities are compared: one as a base reference and one as a variable. Here, the ideal observer has to decide which of the two stimuli is more numerous, by applying the classifier illustrated above.

The probability for the decoder to classify the variable stimulus as the more numerous, is a monotonic function of the variable's stimulus numerosity, well approximated by the integral of a cumulative gaussian function, as described in the Probit model [49]:

$$F(N, \bar{N}, \sigma) = \frac{1}{\sigma\sqrt{2\pi}} \int_{-\infty}^{N} \exp\left(-\frac{(t-\bar{N})^2}{2\sigma^2}\right) dt \,. \tag{4}$$

The median of the Probit indicates the point of subjective equality (where the two stimuli are deemed as having the same numerosity). The curve slope indicates instead the noiseness of the classification. Human-observers psychometric functions for numerosity classification are shallower for larger numerosity in linear scale, having the same width in logarithmic scale: a hallmark of Weber's law. Here we observe the same phenomenon in the output of the classifier (see Fig 5).

## Results

In Fig 5 we present the results of the estimation protocol applied to the magnetisation signal for a system with $M = 18$ spins where $N = 1 - 9$ spin flips occurred with random amplitudes at random times and locations. In Fig 5a we present an example trajectory of the magnetisation for the case with $N = 4$ where we can observe how excitations of varying intensity appear in the system at specific times. In contrast, an average over trajectories of the magnetisation renders no spatial features, Fig 5b. The average magnetisation simply drifts toward overall larger values as the stimuli are added uniformly distributed on average. This highlights the importance of processing the individual trajectories and not their averages. From the individual time signals, we can compose the average spectrum presented in Fig 5c.

The error in numerosity estimation is obtained from the fits of the psychometric functions as described in the Estimation Protocol section. The corresponding fits are displayed in Fig 5d, from smaller to larger numerosities. In Fig 5e, we provide the associated standard deviations of the estimator for the previous case, corresponding to spin flips of random amplitude (orange, random rotations (RR)), and compare them with the case where the spin flips had random rotations but their summed amplitude was constant and equal to $3\pi$ (blue, constant energy (CE)). This comparison is an important test, as many classical systems fail to decode numerosity when the overall excitation amplitude is kept constant. Both cases reproduce the predicted dependence and Weber's law. The case with the constant total amplitude however, is seen to perform worse, due to the small average amplitude of each of the spin flips. Finally, the Weber fraction associated to the results is displayed in the right panel of Fig 5f observing the expected flat dependence. Given the system-size limitations, we were not able to provide signatures for $N > M/2 = 9$ elements. Note that the time window during which we store the time signal is not relevant, as demonstrated in the S1 Appendix, Fig 9. In this figure, we present the power spectrum for the same system, with varying time windows. We observe no qualitative differences in the spectra. In S1 Appendix, we justify the specific numerical values that we chose to ensure consistency over this time window.

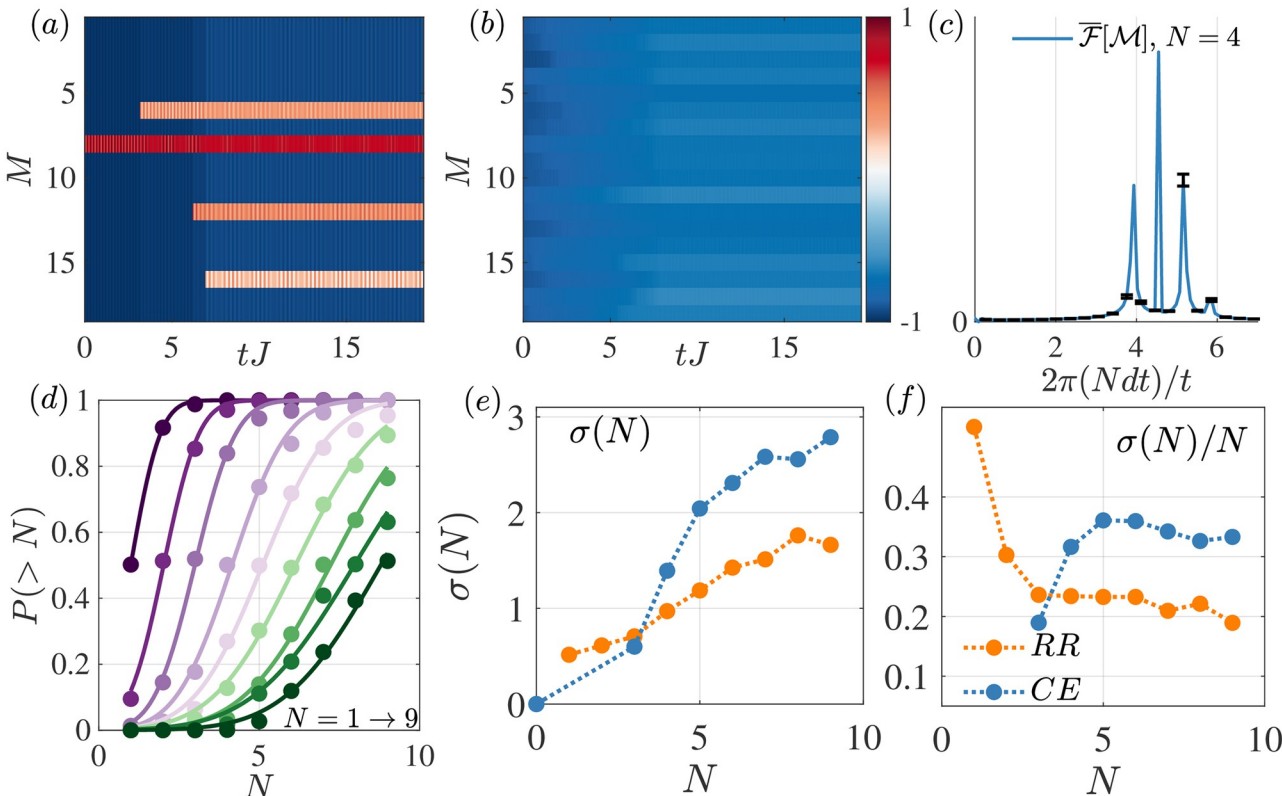

**Fig 5. Numerosity estimation in the quantum spin model.** (a) Example magnetisation profile from one of the sampled trajectories for the case of $M = 18$ spins and $N = 4$ random spin flips, with $\Delta_0 = 0$, $\gamma_l = 0$. (b) Average magnetisation for the same system for a set of $N_t = 200$ trajectories. As the events have random location and intensity, there is no information in the averaged time signal. (c) Average of the amplitude spectrum of the magnetisation signals in the time window between $t = (8J, 18 + \sqrt{2}J)$: the spectra were computed over individual trajectories and sites, then spaced averaged, and finally trajectory averaged. Here we display only the positive frequencies. (d) Psychometric function results for the quantum spin model with $M = 18$, and same parameters as in the top panel, where $N$ spin flips random in location, time, and rotation angle, have occurred before registering the magnetisation. (e) Corresponding standard deviation estimates from the fits of the psychometric functions on the left. Here, we compare the previous case where each event is a spin flip of random amplitude (orange) with a case where the rotations where again random but their sum was constrained to be equal to $3\pi$ (blue). (f) Weber fraction associated to the results in the middle panel.

In order to verify that the essential features of this behaviour in Fig 5d–5f lies in the evolution of the magnetization of the quantum system, we performed a series of validations of our decoding procedure aimed at demonstrating that the decoder and the data fitting do not introduce significant noise components. This analysis is included in Fig 6.

The first test, in Fig 6a, was related to the decoding performance when the libraries were built with a limited number of trajectories. For this we consider the same data set as in Fig 5 in the case of random amplitude of excitations (RR) where we used $N_t = 200$ trajectories. We show that this result is largely independent from the number of trajectories used to construct the templates displaying stably traits of Weber's law using as few as $N_t = 5$ trajectories. This indicates that our ideal decoder performs well with limited samples and that our results are not affected by the specific parameters of the decoder.

Secondly, we test whether the decoder can faithfully capture a pattern of noise present in the signal. To this end, we move away from the data of our quantum simulation and test the decoder with sinusoidal activities computing the spectra from a simple sequence of activations for a noise-free version and one corrupted by $1/f$ noise, proportional to the inverse of the

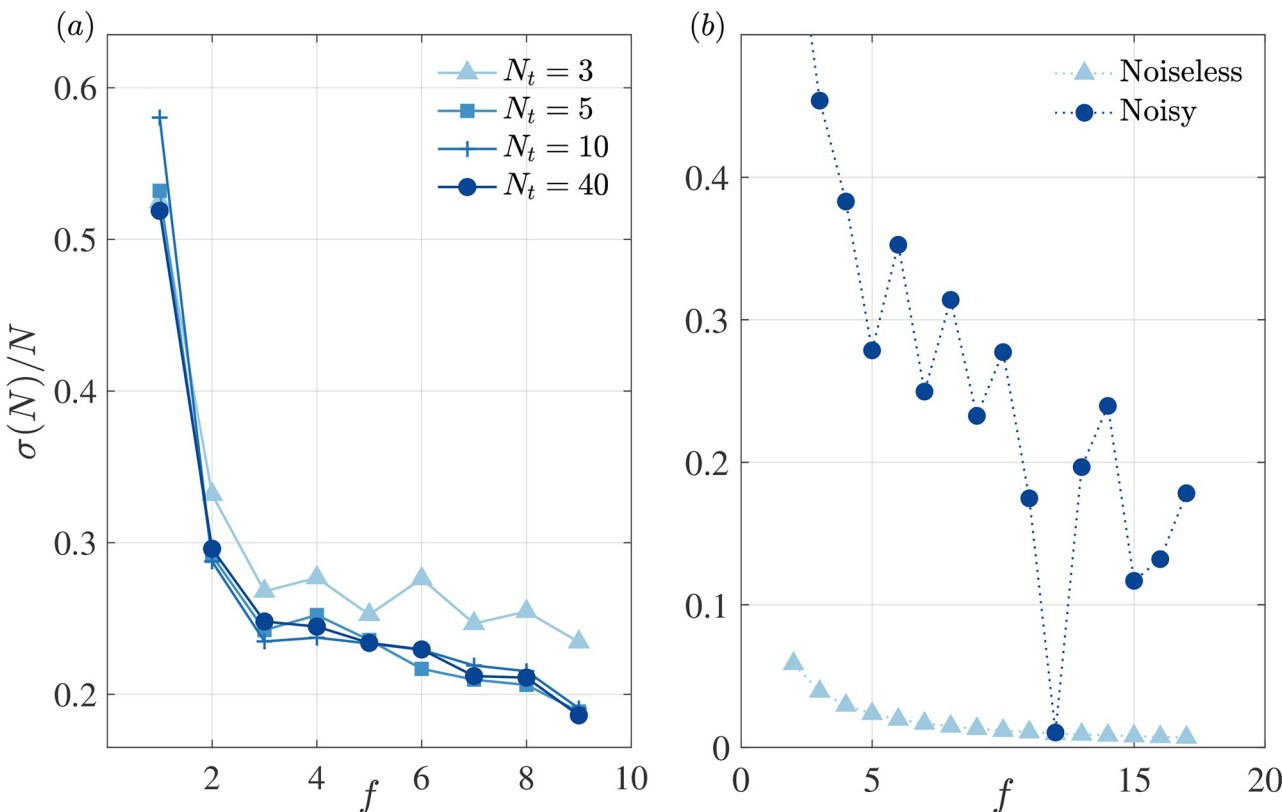

**Fig 6. Decoder validation.** (a) Weber fraction associated to the decoding of numerosity for the magnetisation signals with random amplitude excitations (RR) in Fig 5, for varying number of trajectories $N_t$ = 3, 5, 10, 40 with which we construct our library templates. We observe that the decoding converges with a very limited number of samples. (b) Weber fraction associated to the frequency decoding for sinusoidal activation maps with frequency ranging from $f$ = 1 − 18, both noiseless and with noise proportional to the inverse of the frequency. We show that the decoder does not incorporate additional noise to the analysis by comparing the performance of both classical signals.

sinusoid frequency. More specifically we generated a set of 18 different sine gratings, that differ in temporal frequency with simultaneous activity in all the nodes. The spectra for each of the 18 frequencies contain only one peak and cover nicely the frequency range of the system. We use the decoder to classify the frequency of the sinusoid, in a similar fashion to classifying the number of events, which we did with the quantum system. Fig 6b shows that in the absence of noise in the activity gratings, we obtain good performance with a very low noise floor. Instead, $1/f$ noise, which is more disruptive at low frequencies, renders an output with similar behaviour to shot-noise matching the noise characteristics of the signal. This two additional tests confirm that the decoder does not introduce itself specific noise patterns but simply reveals the intrinsic variability in the activity of the system that is being decoded.

## Discussion

In this work, we have used a quantum spin model to tackle an open problem in human sensory systems, which is how humans are capable of perceiving the numerosity of events. Our results have shown that the time evolution of a very simple fully-connected spin-1/2 Heisenberg model can encode the number of stimuli that it has been exposed to. Moreover, we have shown that its performance is not driven by the overall input energy, that has been normalised,

with the system remaining able to encode numerosity also in the presence of high input energy noise.

It is important to stress that here we have used quantum physics as a statistical tool for information processing, having no implications about the presence of quantum phenomena in the process of numerosity perception in our brain. We have presented a proof-of-principle model where, by adhering to the statistical rules of quantum mechanics instead than the classical ones, we are able to reproduce the behavior of complex networks with a minimal model which takes advantage of the properties of quantum systems. While we do not claim that quantum microscopic interactions in the brain are relevant for sensory processes, we suggest that the powerful mathematical structures existing in quantum models may be useful to simulate complex brain network dynamics.

From the perspective of quantum modeling, there is a wide range of extensions that would be desirable. This would include the use of larger networks by means of matrix product states [50], the use of heterogenous network topologies, e.g. with several fully-connected systems sparsely connected among themselves, or the use of non-Markovian open quantum systems allowing for information back-flow into our spin system. However, we believe that the current iteration of the model represents the essential ingredients for a successful representation of numerosity perception while adhering to minimal complexity.

Importantly, our model is able to exceed previous models in the field of perception of numerosity in two important aspects. Firstly, to our knowledge, no current model exists of how numerosity can be sensed and integrated both in space and time. Some models for numerosity of simultaneously presented items do exist [13–15] and are successful to some degree. However, no models exist for temporal numerosity. This is a notable shortcoming of current literature, as perception of temporal numerosity is strictly linked to spatial numerosity, and both aspects should be conceived as two facets of a single procedure for numerosity processing [51]. Current models possibly can be upgraded to process items presented over time, introducing an ad hoc memory component. However this is a specific solution that cannot be generalized. Instead, our model captures both by construction.

The second interesting issue is that numerosity is encoded in the frequency of the magnetization arising after multiple spin flips and only for the all-to-all connectivity. Thus, our model highlights that in order to measure the number of events unfolding over time, collective interactions measured at all nodes and times may be critical. At least, this is the case for our quantum model. This approach is shifting the focus away from the isolated unit excitation to global network dynamics. Interestingly, global network dynamics measured as endogenous oscillations are an active area of research, and demonstrate the strong potential role in selecting salient information. In this respect our model offers a true paradigm shift, indicating that the frequency of the oscillations may encode the complexity of an incoming stimulus.

In addition, our model is not only capable of encoding numerosity, but also reproduces a critical hallmark of human sensory systems: Weber's law. Whereas Weber's law is a ubiquitous trait of sensory systems, it is particularly challenging from a modelling perspective, since neural firing generally follows Poisson statistics. In our system, Weber's law holds both for randomly rotated systems and even for those where the overall energy is constant despite the change in the number of excitations. We also remark that in our system we have no ad hoc noise sources. The noise in our model mainly arises from the randomness in space, time, and intensity of the local excitations. The randomness-driven noise and the all-to-all connectivity seem to be able to capture Weber's law.

We paired the current evolution of magnetizations with a specific decoding scheme borrowed from analysis and modelling of brain activity [48]. While this model is appealing for its simplicity and biological plausibility, simulating an ideal observer performance, we stress that

it is not an integral part of the quantum model. It constitutes a tool that is used to validate the quantum network model and to demonstrate both its performance and its adherence to Weber's law. Having demonstrated the capability of the network to encode numerosity, any other decoder to extract the correct numerosity may be used, the simplest one being the detection in frequency of peak excitation. Our results have also demonstrated that Weber's Law behaviour is not a consequence of the supervised classifier decoder or the noisiness, but it lies in the magnetization profiles themselves.

The present work can be extended in several ways. For instance, human studies show that visual judgments of numerosity can incorporate biases provided by non-numerical features, such as area, density and energy of the input [52, 53]. While the effects of area and density are often small [6], they can still be important test benches for models of numerosity. Interestingly, in a recent work, Testolin et al. [4] have demonstrated that deep neural networks that have not completed the training, simulate the cues induced by non-numerical features of human perception. In the current quantum simulations we eliminate the non-numerical feature, like total energy, by normalizing the input energy or by adding important energy noise. However, the other two important cues, i.e. area and density, could not be benchmarked adequately given the limitation of the maximum number of excitations allowed (half the length of the network). To demonstrate how these non-numerical features influence the quantum model, one would require substantially larger networks that make use of approximate methods for their simulation, see e.g. matrix product state methods [50].

In addition, in the present work we discussed only two extreme quantum architectures, with either local connectivity or all-to-all connectivity. Here, the all-to-all connectivity, which is the one that encodes numerosity, being closer to biological cortical connectivity. We note, however, that in the cortex the weight of the neuronal connectivity is not equal and can vary dynamically. As a result, it would be interesting to test, with a dedicated set of simulations, whether intermediate architectures could capture the input numerosity, and if those were also sensitive to the modulation of the non-numerical features in the numerosity encoding. In this manner, we could establish a comparison with the results of [4] which show that NN are prone to the influence of non-numerical features.

## Conclusion

All in all, our quantum spin model hinges on very general and minimal assumptions and, as highlighted in the S1 Appendix, remains robust against tuning of system parameters while being able to reproduce essential traits of numerosity perception. We have shown how, after appropriate agnostic decoding, the behaviour of global time-dependent signals conveys information of the number of events the system has experience in the recent past. Refinements of the model can be introduced with sustainable efforts, both to probe larger systems and to investigate phenomena different from numerosity perception. Along these lines, we believe that our model opens unexplored avenues in simulating and interpreting the behaviour of complex brain networks and as a new framework to understand information processing in such systems.

## Supporting information

**S1 Appendix. Discussion on Figs 8 and 9.**
(PDF)

## Author Contributions

**Conceptualization:** Jorge Yago Malo, Guido Marco Cicchini, Maria Concetta Morrone, Maria Luisa Chiofalo.

**Data curation:** Jorge Yago Malo.

**Formal analysis:** Jorge Yago Malo, Guido Marco Cicchini.

**Investigation:** Jorge Yago Malo, Guido Marco Cicchini.

**Methodology:** Jorge Yago Malo, Guido Marco Cicchini, Maria Concetta Morrone, Maria Luisa Chiofalo.

**Supervision:** Maria Concetta Morrone, Maria Luisa Chiofalo.

**Validation:** Maria Concetta Morrone, Maria Luisa Chiofalo.

**Visualization:** Jorge Yago Malo.

**Writing – original draft:** Jorge Yago Malo, Guido Marco Cicchini.

**Writing – review & editing:** Jorge Yago Malo, Maria Concetta Morrone, Maria Luisa Chiofalo.

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
