## [Decision Letter · Decision Letter 0]

19 Jan 2023

PONE-D-22-34934Quantum spin models for numerosity perceptionPLOS ONE

Dear Dr. Yago Malo,

Thank you for submitting your manuscript to PLOS ONE. After careful consideration, we feel that it has merit but does not fully meet PLOS ONE’s publication criteria as it currently stands. Therefore, we invite you to submit a revised version of the manuscript that addresses the points raised during the review process.

We look forward to receiving your revised manuscript.

Kind regards,

Salvatore Lorenzo

Academic Editor

PLOS ONE

Journal Requirements:

"This work was supported by non-commercial basic research grants."

"JYM was supported by the European Social Fund REACT EU through the Italian national program PON 2014-2020, DM MUR 1062/2021. GMC was supported by Horizon 2020 European Research Council Advanced Grant GenPercept No. 832813 (to D.C.B.), Italian Ministry of Education PRIN2017 Grants 2017SBCPZY, and FLAG-ERA Joint Transnational Call 2019 Grant DOMINO. MCM was supported by GenPercept ERC-adv no. 832813 and PRIN 2017. Finally, MC thanks the MIT-UNIPI Project and was funded by European Union - Next Generation EU through the ”Centro Nazionale per Simulazioni, calcolo e analisi dei dati ad alte prestazioni” by the SPOKE on QC for the PNRR."

"This work was supported by non-commercial basic research grants."

6. Please upload a copy of Supporting Information Figure/Table/etc. which you refer to in your text on page 12.

Reviewers' comments:

Reviewer's Responses to Questions

**Comments to the Author**

1. Is the manuscript technically sound, and do the data support the conclusions?

Reviewer #1: Partly

2. Has the statistical analysis been performed appropriately and rigorously? 

Reviewer #1: Yes

3. Have the authors made all data underlying the findings in their manuscript fully available?

Reviewer #1: Yes

4. Is the manuscript presented in an intelligible fashion and written in standard English?

Reviewer #1: Yes

5. Review Comments to the Author

Reviewer #1: SUMMARY

In this paper, the authors introduce an original processing model based on quantum-spin dynamics that can simulate some phenomena related to numerosity perception, such as global sensitivity to numerical information and adherence to Weber’s law in numerosity estimation. The proposed model can (approximately) encode the number of stimuli that it has been exposed to as peaks in its frequency spectrum, suggesting that representations of numerosity information might emerge from relatively simple processing architectures.

COMMENTS

I enjoyed reading this manuscript, which is well-written and well organized. The text is concise and successfully highlights the main aspects of interests of the research work. The results are clearly presented and discussed, and overall the findings appear aligned with the hypotheses proposed by the authors. I also appreciate the interdisciplinary nature of this contribution.

Still, I have a few major concerns that should be addressed to make the paper stronger:

1. The authors argue that current modelling literature has struggled to provide a simple architecture carrying out numerosity perception. However, several recent models have actually shown that numerosity perception could indeed emerge following unsupervised learning in generative neural networks (for a survey, see [1]). Such architectures are much simpler than the CNN-based models mentioned by the authors and are grounded on the theory of statistical mechanics: the building blocks are often energy-based networks derived from spin-glass models, whose dynamics is governed by a Hamiltonian similar to the one described by Eq. 1 in the present work (for theoretical introductions, see [2], [3]). Furthermore, a distinguishing feature of such modeling approach (that is lacking in the present proposal) is that it can explain how network plasticity (i.e., learning) could improve number acuity over time [4]. The authors should acknowledge and discuss these computational models of numerosity perception in their manuscript.

2. Many scientists have recently questioned the existence of a “number sense”, proposing instead that numerosity estimation is carried out by integrating non-numerical magnitudes (e.g., [5], [6]). It would be important to discuss how the proposed modeling framework could account for the influence of non-numerical cues, and possibly show that model responses are based on numerosity rather than other magnitudes embedded in the input signal. Notably, the modeling framework mentioned above has also been successfully used to simulate the influence of visual cues in numerosity perception [7], so I encourage the authors to consider this when further elaborating on such aspects.

3. The authors argue that their framework is not supervised, but the decoding procedure described at pg. 7 assumes that the library of previously learned power spectra are labeled (“associated with a given numerosity”). I think this point should be clarified or rephrased.

4. The authors should better discuss why the proposed “all-to-all” connectivity could be considered a plausible topology for carrying out numerosity perception, and whether such architecture is supported by neurobiological findings.

5. It is shown that the model dynamics is robust to manipulations of the time window. I did not fully understand whether the time window could play a role in simulating reaction times, or whether the time required for the accumulation of numerosity information is constant regardless of the chosen time window. It would be interesting to further comment on this aspect.

MINOR COMMENTS

- Lines 38-39: bad phrasing

- Line 68: “has been” -> “have been”

References

[1] M. Zorzi and A. Testolin, “An emergentist perspective on the origin of number sense,” Philos. Trans. R. Soc. B Biol. Sci., vol. 373, no. 1740, 2018

[2] D. Ackley, G. E. Hinton, and T. J. Sejnowski, “A learning algorithm for Boltzmann machines,” Cogn. Sci., vol. 9, no. 1, pp. 147–169, 1985

[3] E. Agliari, A. Barra, A. Galluzzi, D. Tantari, and F. Tavani, “A walk in the statistical mechanical formulation of neural networks,” in Proceedings of the International Joint Conference on Computational Intelligence, 2014

[4] A. Testolin, W. Y. Zou, and J. L. McClelland, “Numerosity discrimination in deep neural networks: Initial competence, developmental refinement and experience statistics,” Dev. Sci., vol. 23, no. 5, Sep. 2020

[5] T. Leibovich, N. Katzin, M. Harel, and A. Henik, “From ‘sense of number’ to ‘sense of magnitude’ - The role of continuous magnitudes in numerical cognition,” Behav. Brain Sci., vol. 164, 2017

[6] T. Gebuis and B. Reynvoet, “The interplay between nonsymbolic number and its continuous visual properties.,” J. Exp. Psychol. Gen., vol. 141, no. 4, pp. 642–8, Nov. 2012

[7] A. Testolin, S. Dolfi, M. Rochus, and M. Zorzi, “Visual sense of number vs. sense of magnitude in humans and machines,” Sci. Rep., vol. 10, no. 1, pp. 1–13, 2020

6. PLOS authors have the option to publish the peer review history of their article (what does this mean?). If published, this will include your full peer review and any attached files.

Reviewer #1: No

---

## [Author Response · Author response to Decision Letter 0]

22 Mar 2023

Dear editor and reviewer,

We appreciate the feedback and criticism to our manuscript. We include a detailed response to each individual comment in the file "Response to Reviewers" where we also discuss the corresponding changes in the manuscript. We believe that by addressing these points we have strengthen the message and clarity of the paper.

---

## [Editor Report · Decision Letter 1]

5 Apr 2023

Quantum spin models for numerosity perception

PONE-D-22-34934R1

Dear Dr. Yago,

We’re pleased to inform you that your manuscript has been judged scientifically suitable for publication and will be formally accepted for publication once it meets all outstanding technical requirements.

Kind regards,

Salvatore Lorenzo

Academic Editor

PLOS ONE
---

## [Editor Report · Acceptance letter]

13 Apr 2023

PONE-D-22-34934R1 

Quantum spin models for numerosity perception 

Dear Dr. Yago Malo:

I'm pleased to inform you that your manuscript has been deemed suitable for publication in PLOS ONE. Congratulations! Your manuscript is now with our production department. 

Kind regards, 

on behalf of

Dr. Salvatore Lorenzo 

Academic Editor

PLOS ONE